# Mineral Nitrogen Release Patterns in Various Soil and Texture Types and the Impact of Urea and Coated Urea Potassium Humate on Barley Biomass

**Samar Swify** [1,2,*] , **Romas Mažeika** [1] and **Jonas Volungevičius** [1,3]

[1]   Lithuanian Research Centre for Agriculture and Forestry, Instituto al.1, 58344 Akademija, Lithuania; romas.mazeika@lammc.lt (R.M.); jonas.volungevicius@lammc.lt (J.V.)
[2]   Soil and Water Department, Faculty of Agriculture, New Valley University, El-Kharga 72511, Egypt
[3]   Institute of Geosciences, Faculty of Chemistry and Geosciences, Vilnius University, M. K. Ciurlionio g. 21, 01513 Vilnius, Lithuania
[*]   Correspondence: samar.swify@lammc.lt

**Abstract:** Controlled-release urea fertilization is an innovative approach and effective means to reduce the loss of nitrogen and enhance fertilizer use efficiency to optimize crop yield while minimizing the environmental impact. The objective of this study was to investigate the dynamic process of mineral nitrogen (Nmin) release in Luvisols, Cambisols, Retisols, and Arenosols to understand the interaction between soil characteristics and mineral nitrogen release and evaluate the impact of conventional urea compared to coated urea potassium humate on barley biomass production. A pot experiment was conducted under glasshouse conditions ($20 \pm 2$ °C and 60% Humidity). Five treatments including no fertilization as a control (C), 100 kg·N·ha$^{-1}$ of urea (U100), 200 kg·N·ha$^{-1}$ of urea (U200), 100 kg·N·ha$^{-1}$ of coated urea potassium humate (UPH100), and 200 kg·N·ha$^{-1}$ of coated urea potassium humate (UPH200) were applied to four soil and texture types. Our findings indicate that there are different patterns of mineral nitrogen release across the different soil and texture types. Ammonium levels reached their peak point in all soils within 2–7 days after application. On the other hand, the concentration of nitrate $NO_3{}^{-}$–N showed a linear increase over 45 days during the experiment. The Retisol, which had a sandy clay texture, obtained the highest concentration of mineral nitrogen in both forms ($NH_4$ and $NO_3$), while the sandy texture of Arenosol showed the lowest accumulation of mineral nitrogen and its forms. The application of potassium humate caused a delay of 1–4 days in the peak of soil ammonium, which at peak accounted for approximately 25–44% of the mineral nitrogen in the soil. Furthermore, the application of urea and coated urea potassium humate exhibited significant effects on barley biomass with an increase of approximately 14–91% compared to the unfertilized treatment (control). This research contributes to our understanding of nutrient dynamics in diverse soil environments and provides insights into optimizing sustainable fertilization strategies such as controlled-release fertilizer application. The implications of these findings highlight the significance of tailored nutrient management practices based on soil texture type, which can lead to improved agricultural productivity and environmental impact.

**Keywords:** mineral nitrogen release; urea; soil texture type; potassium humate; coated; barley biomass; nutrient management

## 1. Introduction

Nitrogen is a crucial element for barley growth and serves as a fundamental component in various metabolic activities, encompassing the synthesis of proteins and the production of chlorophyll [1,2]. Urea is one of the most prevalent nitrogenous fertilizers and accounts for approximately 43% of global nitrogen fertilizer sales [3,4]. Urea is extensively utilized in the agricultural sector owing to its high nitrogen concentration and cost-effectiveness [5].

However, the efficiency of urea as a main nitrogen source can be influenced by several factors, including soil characteristics, microbial activity, and environmental conditions [6–9]. In commonly applied fertilizers such as urea, the predicted efficiency is around 30–70% of nitrogen application, but usually, the efficiency of urea is lower than 50% [10–13]. The nitrogen loss could cause ecological contamination via tainting groundwater by means of nitrate leaching and by adding to the emission of greenhouse gases into the atmosphere, with reports of up to 160 $kg \cdot N \cdot ha^{-1} \cdot yr^{-1}$ lost to leaching and up to 143 $kg \cdot N \cdot ha^{-1} \cdot yr^{-1}$ lost as greenhouse gases, which is a matter of great concern [13–15]. Therefore, it is essential to improve the use efficiency of urea fertilizer to reduce nitrogen loss [13,16,17].

The efficiency of urea-N fertilizer application can be enhanced by ensuring a balance between N supply and crop N demand, in conjunction with the minimization of N losses [13,18]. The utilization of controlled-release urea fertilizers (CRUF) is one strategy to minimize nitrogen losses and improve the uptake efficiency of the fertilizer [19–22]. Controlled-release urea fertilizers such as coated urea, which are intended to coordinate their N release pattern with crop N consumption rates, could potentially be a significant advance in resolving the synchrony issue [18,21,23–25] and reduce the rate of nitrogen loss to the environment [24,26–28].

Moreover, the application of CRUF is also a secure option for minimizing the damage to crop germination and increasing plant biomass [19,20,29–31]. The combination of inorganic fertilizers and organic materials by coating or granulation is progressively gaining popularity owing to its potential advantages on crop yields and soil health when compared to the use of organic material or inorganic fertilizer as the only source of nutrients [13,32–34]. Potassium humate, derived from organic matter, is known to enhance soil fertility and nutrient availability [35,36].

Humic substances are a type of macromolecular natural organic matter that comprises a polyelectrolyte with a negative charge [36–38] that comprises a polyelectrolyte with a negative charge [36–38]. This polyelectrolyte is characterized by the presence of a hydrophobic core and a diverse array of functional groups such as quinone, aldehyde, carboxyl, phenolic, and alcoholic hydroxyls, as well as ether [35,36,39–41]. Moreover, humic acid contains unbound carboxylic and amide groups, which demonstrate chelating capability, dispersibility, and adsorption when humic acid is used as a polymer [37,38]. This makes it suitable for use as a control-release agent and absorption enhancer in fertilizers [37,42]. Coating urea with humic acid can effectively slow down the release of urea by inhibiting the processes of ammonification and nitrification [32,37]. Therefore, the conversion rate of urea-N slowed due to the chemical reactions that occur between humic substances and urea, respectively. This is due to the bonding of urea-N with amino, carboxyl, and hydroxyl groups in humic molecules [37].

According to previous studies, it has been reported that combining urea with potassium humate can improve nitrogen use efficiency and mitigate losses through volatilization or leaching [35,39,40,43,44]. The application of a combination of urea with humic compounds enhances nitrogen transformation to the ammonium form $NH_4^+$–N rather than the ammonia form $NH_3$, which would be expected to reduce environmental damage [22,34,39,40]. The higher bond strength with higher concentrations of humate can slow the release of urea into the ammonium form, which then also slows its conversion into the nitrate form, leading to lower N losses [34,37,39,41].

Moreover, the presence of humic acid can effectively impede the action of urease and thus decrease the rate of urea hydrolysis [45–47]. This phenomenon ultimately results in a diminished concentration of ammonium ($NH_4^+$–N) in the soil, which consequently mitigates the hazards of ammonia ($NH_3$) volatilization and potential nitrification, leading to a significant reduction in N losses [22,45]. Humic acid has also exhibited advantageous properties in regulating soil acidification [10,48,49].

Furthermore, the application of urea potassium humate has been observed to stimulate the growth and development of roots, thereby resulting in a significant increase in root

biomass [50–52]. The humic-acid application also can increase soil water retention and reduce the inorganic fertilizer's solubility [53].

Coating urea with humate leads to a stable product [39] with controlled nitrogen release [39–41]. However, the effectiveness of this combination may vary depending on the soil type [19,54]. Different soils possess distinct physicochemical properties and microbial populations, which can influence nutrient availability and plant response [8,19,54,55].

Hence, the present investigation endeavors to (1) evaluate the patterns of mineral nitrogen release in diverse soil types, and (2) appraise the impact of urea and coated urea potassium humate on the production of barley biomass. By quantifying these aspects, we can further our understanding of the nitrogen dynamics in different soil types following application with uncoated and coated urea in different soil types and supply valuable information concerning nutrient management.

## 2. Materials and Methods

### 2.1. Nitrogen Sources

This research was conducted to compare urea as the main nitrogen fertilizer with coated urea potassium humate (UPH) as a modified urea compound fertilizer (coated form) in a pot experiment under glasshouse conditions (20 ± 2 °C and 60% Humidity). The nitrogen sources that were used (Figure 1) included Urea, which contains 46.2% as a total N, and coated urea potassium humate (UPH), which contains 40% N and 1% Potassium Humate; these were produced by AB Achema Company, Kaunas, Lithuania.

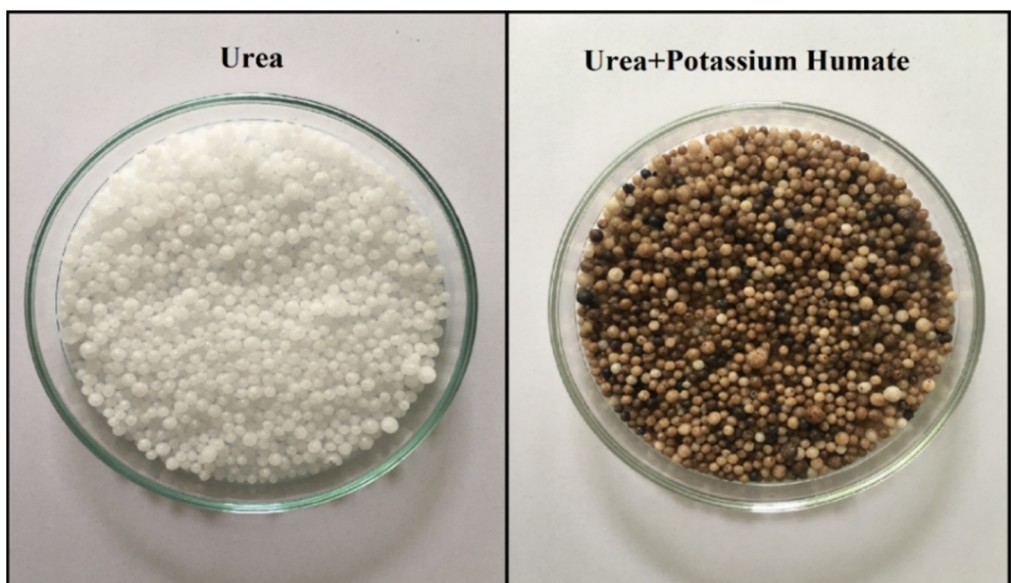

**Figure 1.** Nitrogen sources included urea granules and coated urea potassium humate granules.

### 2.2. Study Areas and Soil Characteristics

The study areas as shown in Figure 2 (in the Republic of Lithuania territory) are located in the mid-latitude, transitional climate zone. The soils are formed and developed under wet (600–820 mm) and cold (6.0–7.5 °C) climate conditions [56,57]. Four soil types used in our study were collected from Lithuania's agricultural soils. Soil profiles were classified and named according to the WRB 2022 classification [58]. The first soil was collected from the Rumokai (Vilkaviškis distr. mun.) research station field (54°41′41.3″ N, 22°59′35.0″ E) of the Lithuanian Research Center of Agriculture and Forestry. The field soil type is Endogleyic Haplic Luvisol (Loamic, Aric) [58], which was chosen to represent the loam texture.

The second type was collected from the Dotnuva (Kėdainiai distr. Mun.) field (55°23′23.8″ N, 23°52′12.9″ E) at the Lithuanian Research Center of Agriculture and Forestry. The soil type was Endogleyic Epistagnic Endocalcaric Cambisol (Loamic, Aric, Drainic) [58]

to represent the sandy loam texture. The third type was collected from Kazlu Ruda municipality (54°45′16.7″ N, 23°29′26.0″ E). The soil is formed in limnoglacial sands and is therefore named Dystric Arenosol (Aric) [58]. The fourth type was collected from Vezaiciai (Klaipėda distr. Mun.) (55°42′41.6″ N, 21°26′25.5″ E), which is the research station field of the Lithuanian Research Center of Agriculture and Forestry. The soil was Glossic Epigleyic Dystric Retisol (Clayic, Aric, Cutanic, Drainic) [58] and it is represented in sandy clay texture. Soil types and their physiochemical characteristics are shown in Table 1.

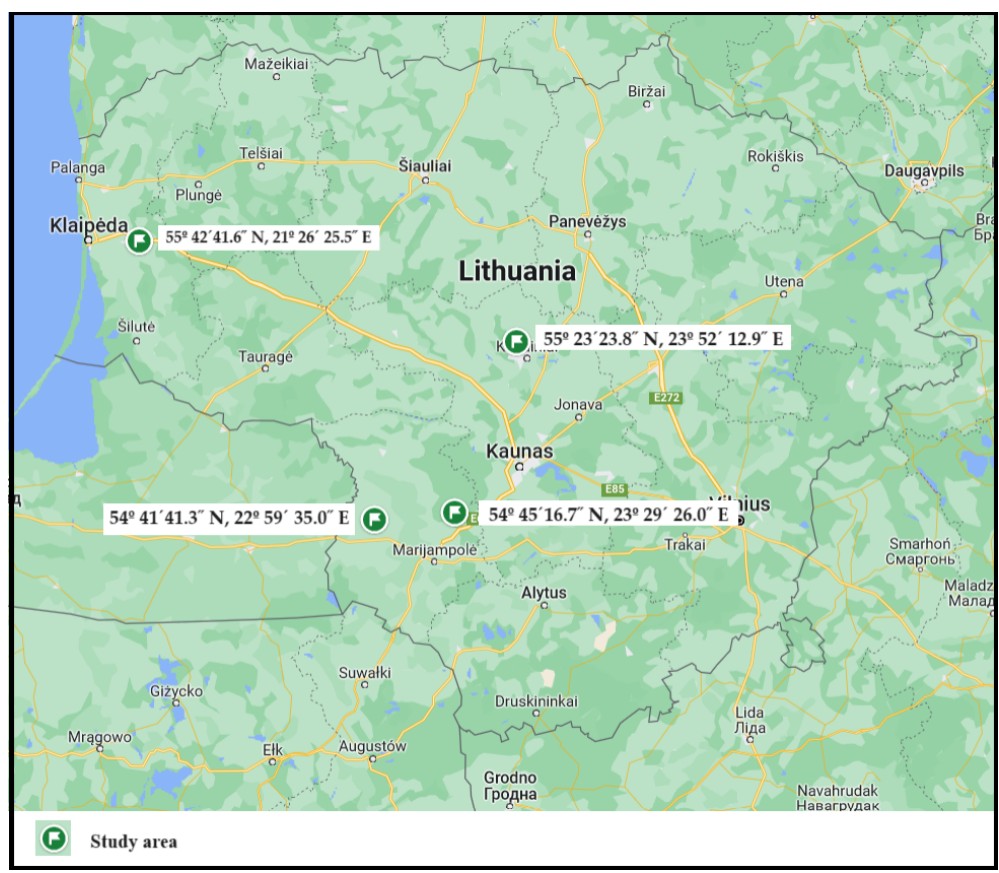

**Figure 2.** The study areas' locations (Source: Google Maps).

**Table 1.** Physiochemical characteristics of different soil types.

| Soil Characteristics | Rumokai | Dotnuva | Kazlu Ruda | Vezaiciai |
|---|---|---|---|---|
| Soil Type | Endogleyic Haplic Luvisol (Loamic, Aric) | Endogleyic Epistagnic Endocalcaric Cambisol (Loamic, Aric, Drainic) | Dystric Arenosol (Aric) | Glossic Epigleyic Dystric Retisol (Clayic, Aric, Cutanic, Drainic) |
| Texture | Loam (L) | Sandy Loam (SL) | Sand (S) | Sandy Clay (SC) |
| $pH_{Kcl}$ mol/L | 7.4 | 6.9 | 5.3 | 4.5 |
| $P_2O_5$ mg·kg$^{-1}$ | 244 | 266 | 102 | 139 |
| $K_2O$ mg·kg$^{-1}$ | 201 | 308 | 40 | 240 |
| SOM * (%) | 1.04 | 1.26 | 0.79 | 1.25 |
| SOC ** (%) | 0.60 | 0.73 | 0.46 | 0.72 |
| $Ec_e$ mS/m | 14.4 | 12.7 | 2.31 | 6.10 |
| Total N (%) | 0.148 | 0.151 | 0.013 | 0.120 |
| Nmin mg·kg$^{-1}$ | 23.99 | 37.32 | 1.74 | 41.60 |
| $NH_4^+$–N mg·kg$^{-1}$ | 2.06 | 1.33 | 0.98 | 3.68 |
| $NO_3^-$–N mg·kg$^{-1}$ | 21.93 | 35.99 | 0.76 | 37.92 |

* SOM = Soil organic matter, ** SOC = Soil organic carbon.

### 2.3. Experimental Design and Treatments

A pot experiment was conducted at the glasshouse of the Lithuanian Research Center for Agriculture and Forestry, Agrobiology Laboratory in 2021 to investigate the effect of different soil and texture types on mineral nitrogen (Nmin) release patterns by using conventional urea and coated urea potassium humate. The experiment was conducted for 45 days by growing spring barley (Hordeum vulgare L.) cultivar "Ema DS". The experimental design included 60 pots set in 4 groups (A, B, C, and D), one for each soil type, and each group consisted of 15 pots as shown in Figure 3. The pot size was 25 cm in diameter and 25 cm in height and was filled with 10 kg of soil packed to a bulk density of approximately 1.3–1.8 Mg m$^{-3}$.

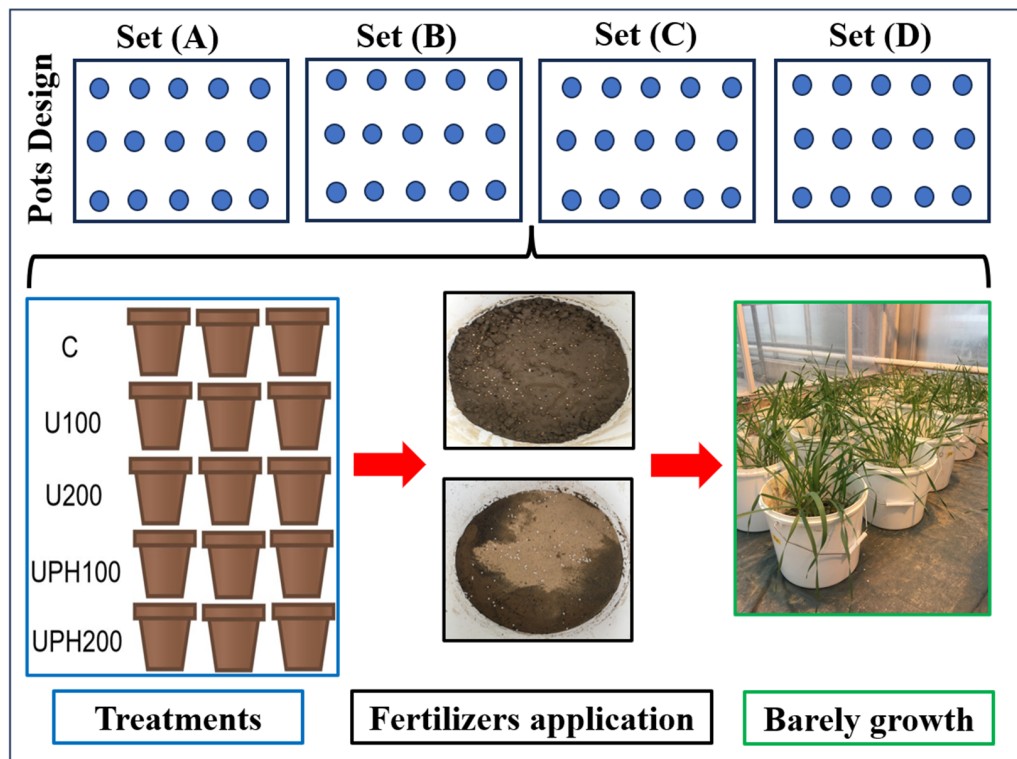

**Figure 3.** The general schema of the pot experiment included four sets of pots: loamy (**A**), sandy loam (**B**), sandy (**C**), and sandy clay (**D**) texture.

The pots were filled with 4 soil texture types (L, SL, S, and SC), and each group was filled with one soil texture type and divided into five treatments with three replicates. Treatments including control (C), urea 100 kg·N·ha$^{-1}$ (U100), urea 200 kg·N·ha$^{-1}$ (U200), coated urea potassium humate 100 kg·N·ha$^{-1}$ (UPH100), and coated urea potassium humate 200 kg·N·ha$^{-1}$ (UPH200) as shown in Figure 3.

The treatments were applied to the surface of the pots and then covered with a thin layer of soil and watered. The pots were maintained in the glasshouse climatic conditions (20 ± 2 °C and 60% Humidity) and irrigated manually according to their requirements. Soil samples were collected to a depth of 10 cm from each pot with a sample size of 100 g, commencing from day 1 of the experiment and subsequently on days 4, 7, 14, 21, 28, and 45, and this sequence of the dates was chosen based on a pre-study to study the release of these materials in incubation tests without plants. Spring barley (*Hordeum vulgare* L.) cultivar "Ema DS" was sown at a depth of 1 cm into the soil at equal intervals of distance. There are no additional mineral fertilizers applied.

### 2.4. Soil Sampling and Laboratory Analysis

Soil for the experiment was collected from the topsoil layer of four profiles at depths of 0–30 cm in order to represent the four various soil types (Table 1). The soil was air-dried and then ground to a size that could pass through a 2 mm sieve. Detailed analysis of the soil properties and mineral nitrogen content was performed by the Agrochemical Research Laboratory at the Lithuanian Research Center for Agriculture and Forestry. Soil pH was determined according to ISO 10390:2005 [59]. Soil-available $K_2O$ and $P_2O_5$ were extracted by using soil suspension 1:20 (wt./vol) of ammonium lactate-acetic acid extraction (pH 3.7) and determined [60]. Soil electrical conductivity (Ece) determination was according to ISO 11265:1994 [61]. Total nitrogen was measured according to ISO 11261-1995 [62,63]. Mineral nitrogen in the top 10 cm of the pots was determined by using a spectrometric flow injection analysis (FIA) method developed by the laboratory, including both nitrate concentration (Sum of $NO_3$–N and $NO_2$–N) and ammonium concentration ($NH_4$–N) [43]. The mineral nitrogen (Nmin) was calculated by adding the sum of nitrate and nitrite nitrogen to ammonium nitrogen. The organic soil carbon content was determined using dry combustion according to ISO 10694:1995 [64]. Barley vegetative material was cut 45 days after sowing and dried at +105 °C.

### 2.5. Statistics Analysis

The statistical analysis software IBM SPSS 25.0 (Armonk, NY, USA) was used for data on mineral N, ammonium $NH_4^+$–N, nitrate $NO_3^-$–N, barley fresh weight, and dry matter. All the data were subjected to analysis of variance (ANOVA), and separate statistical significance by Duncan's multiple range test at the 5% level. Pearson's correlation was performed on the relationship between the time and mineral N release and its forms nitrate $NO_3^-$–N and ammonium $NH_4^+$–N.

## 3. Results

### 3.1. Mineral Nitrogen Patterns in Different Soil and Texture Types

The concentration of mineral nitrogen and its main forms ($NH_4^+$–N and $NO_3^-$–N) exhibited distinct variations and different patterns of release according to the soil and texture type as shown in Figure 4. The concentration of ammonium-N in the top 10 cm of soil in the pots increased rapidly after urea application, reaching a peak after 2–7 days, followed by a gradual decline to near-zero levels in all soils except the Retisol (sandy clay) as in Figure 4. The peak of ammonium concentration varied with soil type, urea source, and rate, and it was highest in the Retisol (sandy clay), followed in sequential order by the Cambisol (sandy loam), Luvisol (loam), and Arenosol (sand). The highest peak concentrations were in the U200 and UPH200 treatments. However, there was no significant linear relationship between ammonium-N concentration and time.

Conversely, the release patterns of $NO_3^-$–N exhibited a degree of variation after a fortnight (the fifth point in the graphs of $NO_3^-$) as shown in Figure 4. The concentration of nitrate $NO_3^-$–N exhibited a highly significant correlation with time across all treatments and soil types (Table 2). The highest peak was found in the Retisol (sandy clay), followed by the Luvisol (loam), Cambisol (sandy loam), and Arenosol (sand). Notably, U200 treatment achieved the highest significant mean of $NO_3^-$–N concentration in Cambisol followed by the treatment of UPH200 in Luvisol. The nitrate-N concentration increased at a linear rate over the 45 days of measurements, without further change until the end of the experiment (Figure 4, Table 2). The total mineral N also increased with time but only achieved significance at the 5% level in 11 of the 16 soil-fertilizer combinations tested (Figure 4, Table 2). At the conclusion of the experiment on day 45, soil in the pots to a depth of 30 cm was analyzed for mineral N. Among the soils (Table 3), the highest mineral N concentrations were in the Retisol (sandy clay), followed by the Cambisol (loam) and Luvisol (sandy loam), while the lowest concentrations were in the Renosol (sand). Among the fertilizer treatments, the highest concentration was in U200 followed by UPH200, UPH100, and U100.

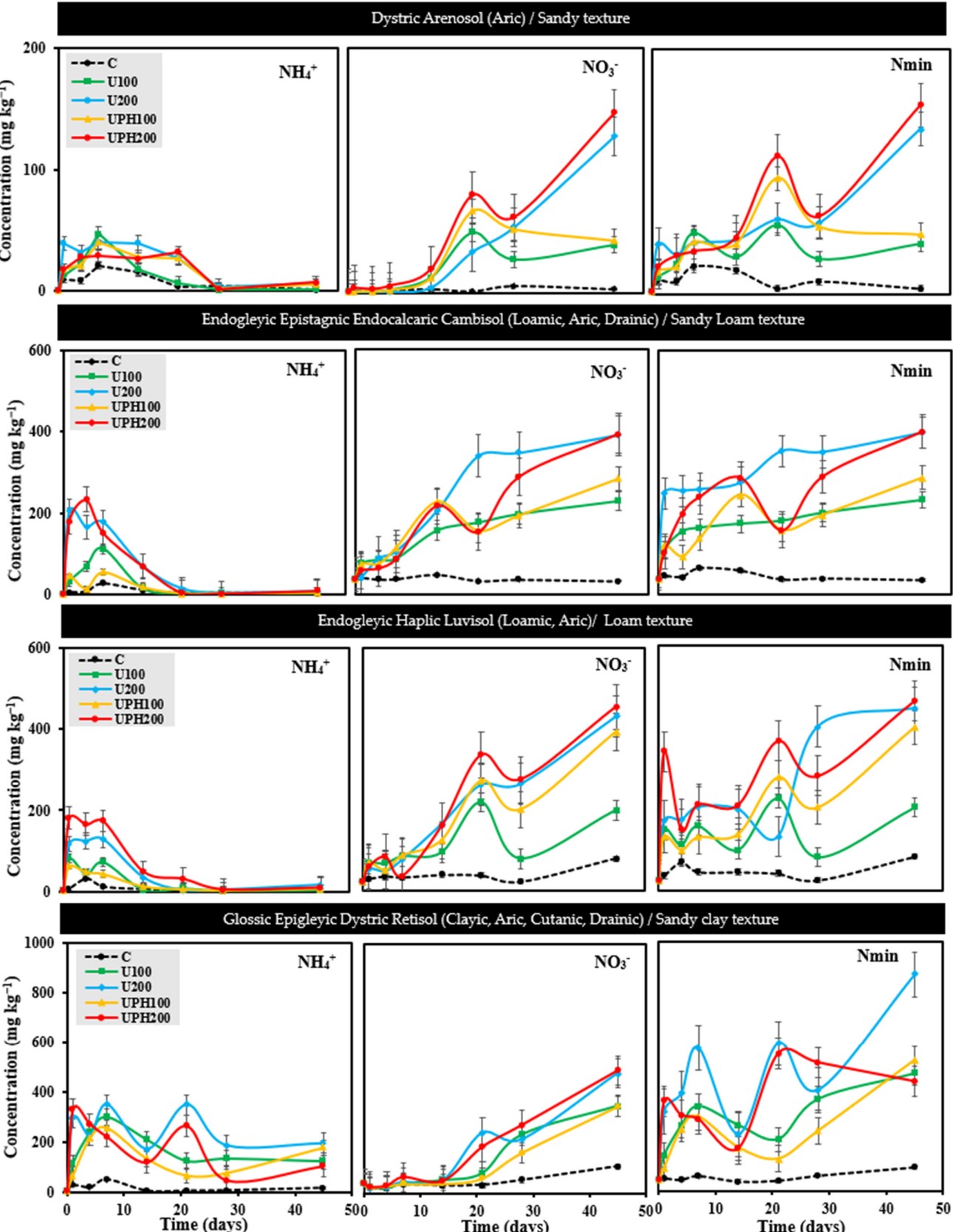

**Figure 4.** The release patterns of mineral nitrogen (Nmin) and its forms ammonium ($NH_4^+$–N) and nitrate ($NO_3^-$–N) as a relation between concentration (mg·kg$^{-1}$) and time (days) in different soil and texture types.

**Table 2.** Pearson's correlation between the time, nitrate ($NO_3^-$–N), and mineral nitrogen (Nmin) release concentration (mg·kg$^{-1}$) in different soil types.

| Treatments | Mean | | Correlation Coefficient | | *p*-Value | |
|---|---|---|---|---|---|---|
| | \multicolumn Dystric Arenosol (Aric)/Sandy Texture | | | | | |
| | $NO_3^-$–N | Nmin | $NO_3^-$–N | Nmin | $NO_3^-$–N | Nmin |
| Control (C) | 2.22 | 10.36 | 0.612 | −0.249 | 0.107 | 0.552 |
| U100 | 16.70 | 30.52 | 0.827 * | 0.519 | 0.011 | 0.188 |
| U200 | 27.98 | 51.68 | 0.953 ** | 0.918 ** | 0.000 | 0.001 |
| UPH100 | 22.29 | 40.32 | 0.770 * | 0.593 | 0.025 | 0.121 |
| UPH200 | 40.24 | 58.23 | 0.961 ** | 0.920 ** | 0.000 | 0.001 |
| | Endogleyic Epistagnic Endocalcaric Cambisol (Loamic, Aric, Drainic)/Sandy loam texture | | | | | |
| | $NO_3^-$–N | Nmin | $NO_3^-$–N | Nmin | $NO_3^-$–N | Nmin |
| Control (C) | 36.88 | 43.57 | −0.432 | −0.422 | 0.285 | 0.298 |
| U100 | 130.46 | 155.84 | 0.948 ** | 0.826 * | 0.000 | 0.012 |
| U200 | 194.21 | 271.25 | 0.942 ** | 0.785 * | 0.000 | 0.021 |
| UPH100 | 144.50 | 158.10 | 0.898 ** | 0.844 ** | 0.002 | 0.008 |
| UPH200 | 161.42 | 212.78 | 0.964 ** | 0.819 * | 0.000 | 0.013 |
| | Endogleyic Haplic Luvisol (Loamic, Aric)/Loamy texture | | | | | |
| | $NO_3^-$–N | Nmin | $NO_3^-$–N | Nmin | $NO_3^-$–N | Nmin |
| Control (C) | 37.80 | 45.95 | 0.740 * | 0.455 | 0.036 | 0.258 |
| U100 | 104.26 | 131.79 | 0.742 * | 0.478 | 0.035 | 0.231 |
| U200 | 165.83 | 219.56 | 0.989 ** | 0.846 ** | 0.000 | 0.008 |
| UPH100 | 152.56 | 175.51 | 0.957 ** | 0.924 ** | 0.000 | 0.001 |
| UPH200 | 178.23 | 255.47 | 0.956 ** | 0.737 * | 0.000 | 0.037 |
| | Glossic Epigleyic Dystric Retisol (Clayic, Aric, Cutanic, Drainic)/Sandy clay texture | | | | | |
| | $NO_3^-$–N | Nmin | $NO_3^-$–N | Nmin | $NO_3^-$–N | Nmin |
| Control (C) | 42.99 | 53.15 | 0.879 ** | 0.753 * | 0.004 | 0.031 |
| U100 | 105.72 | 260.86 | 0.946 ** | 0.793 * | 0.000 | 0.019 |
| U200 | 139.95 | 425.08 | 0.958 ** | 0.750 * | 0.000 | 0.032 |
| UPH100 | 92.28 | 216.81 | 0.922 ** | 0.754 * | 0.001 | 0.031 |
| UPH200 | 144.67 | 331.69 | 0.969 ** | 0.621 | 0.000 | 0.100 |

** Correlation is significant at the 0.01 level (two-tailed). * Correlation is significant at the 0.05 level (two-tailed).

**Table 3.** The effect of the soil texture type and urea fertilizers application on accumulative mineral nitrogen and its forms ($NH_4^+$–N and $NO_3^-$–N) in the surface layer 0–30 cm depth after 45 days.

| Soil Texture Type | $NH_4^+$–N | $NO_3^-$–N | Nmin |
|---|---|---|---|
| | mg·kg$^{-1}$ | | |
| Sand | 16.33 [a] | 21.89 [a] | 38.22 [a] |
| Sandy loam | 43.03 [a] | 133.49 [c] | 168.31 [b] |
| Loam | 37.92 [a] | 127.74 [bc] | 165.66 [b] |
| Sandy clay | 137.55 [b] | 105.12 [bc] | 257.52 [c] |
| *p*-value | *p* < 0.003 | *p* < 0.002 | *p* < 0.001 |
| **Treatments** | | | |
| Control (C) | 9.87 [a] | 29.97 [a] | 38.26 [a] |
| U100 | 56.60 [abc] | 89.29 [b] | 144.75 [b] |
| U200 | 94.84 [c] | 132.00 [d] | 241.89 [d] |
| UPH100 | 45.71 [abc] | 102.91 [bcd] | 147.69 [b] |
| UPH200 | 86.51 [bc] | 131.14 [cd] | 214.54 [bcd] |
| *p*-value | *p* < 0.021 | *p* < 0.003 | *p* < 0.001 |

Numbers followed by the same letter in the same column are not significantly different at the 5% level of Duncan's test.

### 3.2. Impact of Soil and Texture Types on Mineral Nitrogen Accumulation

The accumulative mineral nitrogen (Nmin) released during 45 days of the pot experiment and its forms ($NH_4^+$–N and $NO_3^-$–N) were significantly ($p \leq 0.01$) influenced by the soil texture type (Table 3). The Arenosol with a sandy texture exhibited the lowest accumulative mineral nitrogen, ammonium, and nitrate concentration at means of 38.22, 16.33, and 21.89 mg·kg$^{-1}$, respectively. Otherwise, the highest significant concentration of mineral nitrogen and ammonium was observed in the Retisol with sandy clay texture at 257.52 and 137.55 mg·kg$^{-1}$, respectively. By contrast, the highest significant concentration of nitrate was observed in Cambisols with sandy loam texture at 133.49 mg·kg$^{-1}$.

Furthermore, the urea and coated urea treatments significantly ($p \leq 0.01$) affected the mineral nitrogen, ammonium, and nitrate (Table 3). The U200 treatment recorded the highest significant concentration of the mineral nitrogen at 241.89 mg·kg$^{-1}$ followed by UPH 200 at 214.54 mg·kg$^{-1}$ and UPH100 at 147.69 mg·kg$^{-1}$. Moreover, the application of U200 significantly increased the accumulation of ammonium and nitrate at 94.84 mg·kg$^{-1}$ and 132 mg·kg$^{-1}$, respectively. Treatments of UPH200 and UPH100 showed significantly higher concentrations of nitrate $NO_3^-$–N than U100 as shown in Table 3. By contrast, UPH100 exhibited significantly lower levels of ammonium $NH_4^+$–N concentration than U100.

One of the important factors that could affect the release and accumulation of mineral nitrogen is the soil pH. These soil types, Arenosol, Cambisol, Luvisol, and Retisol, had varied in their pH values as presented in Table 1. Based on the USDA classification of pH [65], the Retisol is a very strongly acidic soil at pH 4.5 followed by Arenosol, which is strongly acidic soil at pH 5.3 [65]. Otherwise, Cambisol and Luvisol are classified as neutral and slightly alkaline soil at 6.9 and 7.4 [65].

The application of urea and coated urea increased the pH of the treated soils after 45 days, but there were no significant differences observed as shown in Table 4. The application of urea and coated urea increased the values of the pH of the Arenosol compared to the control. Otherwise, the application of the urea and coated urea did not affect the pH in Cambisols. Moreover, treating Luvisols with urea and coated urea slightly decreased the pH. However, the application of U200 increased the pH in the Retisol compared to the other treatments. The pH can affect the release of nitrogen by delaying the hydrolysis of urea and inhibiting urease activity as well as quickening the nitrification process [66,67]. The pH of the soil played a vital role in delaying the peak of the ammonium in the Arenosols and Retisols to 7 days compared to the other soil types, which reached the peak in 2–4 days.

**Table 4.** The changes in pH values of the 0–10 cm layer of the pots after the application of urea and coated urea in different soil types.

| The Application | Arenosol (S) | Cambisol (SL) | Luvisol (L) | Retisol (SC) |
|:---:|:---:|:---:|:---:|:---:|
| Before | 5.3 | 6.9 | 7.4 | 4.5 |
| Control | 6.00 [a] | 6.90 [a] | 7.03 [b] | 4.27 [abc] |
| U100 | 6.20 [abc] | 6.93 [a] | 7.03 [b] | 4.03 [a] |
| U200 | 6.63 [c] | 6.83 [a] | 6.93 [ab] | 4.67 [c] |
| UPH100 | 6.37 [abc] | 6.83 [a] | 7.00 [ab] | 4.10 [a] |
| UPH200 | 6.47 [abc] | 6.87 [a] | 6.93 [ab] | 4.33 [abc] |
| *p*-value | 0.16 | 0.29 | 0.68 | 0.11 |

Numbers followed by the same letter in the same column are not significantly different at the 5% level of Duncan's test.

### 3.3. Availability Assessment of Mineral Nitrogen in the Soil

The results of the descriptive statistics exhibit significant fluctuations in the soil contents of mineral nitrogen, which are presented in Table 5. The data reveal that Nmin soil content varied between 10.36 and 8.23 mg·kg$^{-1}$ in Arenosol (sandy), while in Cambisol (sandy loam), it ranged from 43.57 to 271.25 mg·kg$^{-1}$, from 45.95 to 255.47 mg·kg$^{-1}$ in Luvisol (loamy), and in Retisol (sandy clay), it ranged from 53.15 to 425.08 mg·kg$^{-1}$.

**Table 5.** Descriptive statistics of soil mineral nitrogen (Nmin), ammonium, and nitrate of the study soil texture types.

|  | Soil Texture Type | Min. | Max. | Mean | SD | CV (%) |
|---|---|---|---|---|---|---|
| Nmin (mg·kg$^{-1}$) | Sand | 10.36 | 58.23 | 38.22 | 18.85 | 23.41 |
|  | Sandy loam | 43.57 | 271.25 | 168.31 | 84.25 | 104.61 |
|  | Loam | 45.95 | 255.47 | 165.66 | 81.46 | 101.15 |
|  | Sandy clay | 53.15 | 425.08 | 257.52 | 138.76 | 172.29 |
| NH$_4^+$–N (mg·kg$^{-1}$) | Sand | 8.14 | 23.70 | 16.33 | 5.77 | 7.17 |
|  | Sandy loam | 6.69 | 80.96 | 43.03 | 35.26 | 43.78 |
|  | Loam | 8.15 | 77.24 | 37.92 | 27.45 | 34.08 |
|  | Sandy clay | 16.53 | 221.70 | 137.55 | 76.24 | 94.66 |
| NO$_3^-$–N (mg·kg$^{-1}$) | Sand | 2.22 | 40.24 | 21.89 | 14.03 | 17.43 |
|  | Sandy loam | 36.88 | 194.21 | 133.49 | 59.01 | 73.28 |
|  | Loam | 37.80 | 178.23 | 127.74 | 57.57 | 71.48 |
|  | Sandy clay | 42.99 | 144.67 | 105.12 | 41.24 | 51.21 |

The coefficient of variation (CV) is a valuable statistic to compare the degree of variation between different datasets, even if the means differ significantly. A coefficient of variation (CV) of less than 10% indicates low variability, 10–100% suggests moderate variability, and a CV greater than 100% indicates high variability [68]. In this research, the coefficient of variation (CV) of the soil Nmin was determined. The CV of the soil Nmin was found to be 23.41 in the Arenosol with a sandy texture, which indicated that the sandy texture samples had moderate variability of the soil Nmin. Conversely, the Cambisol (sandy loam), Luvisol (loamy), and Retisol (sandy clay) soils had a CV greater than 100%, signifying high variability of mineral nitrogen in these soil samples.

The availability of Nmin in soils can be categorized as Low < 40, Moderate 40–80, and High > 80 mg·kg$^{-1}$ [69]. Based on this classification, the Arenosol (sandy) soil had a low availability of mineral nitrogen at 38.85 mg·kg$^{-1}$. In contrast, the Cambisol (sandy loam), Luvisol (loamy), and Retisol (sandy clay) soils had a high availability of mineral nitrogen at 168.31, 165.66, and 257.52 mg·kg$^{-1}$, respectively.

*3.4. Effect of Urea Compounds on Germination and Barley Biomass*

Germination of the barley seed in the pots commenced on the fourth day following the application of the fertilizers with a rate of 70–90%. The utilization of urea fertilizers impacted the germination of barley significantly ($p \leq 0.01$).

In all the soil texture types, UPH200 and U200 exhibited noticeably lower significant germination rates compared to the control. Conversely, no significant differences were observed between UPH100 and U100 in comparison to the control. The outcome in terms of barley green biomass yield at the conclusion of the 45 days in the pots experiment under the glasshouse conditions is presented in Figure 5.

Furthermore, the utilization of urea and coated urea potassium humate treatments had a significant effect ($p \leq 0.05$) on the barley's green biomass. Among these treatments, UPH200 recorded the highest significant mean of 31.44 g followed by U200 with a similar mean of 31.25 g. Conversely, there was no significant disparity recorded between U100 with a mean of 25.91 g, and UPH100 with a mean of 25.05 g in comparison to the control at 20.47 g. Within the soil type groups, the treatments displayed significant differences from one another as shown in Figure 6.

Notably, within the Arenosol (sandy), the application of these treatments significantly impacted the green biomass of barley ($p \leq 0.01$). The highest significant biomass was obtained under treatment of UPH200 at 16.57 g followed by U200 at 14.71 g. U200 and UPH100 had no differences observed between each other. Similarly, U100 showed no significant difference when contrasted with the control (Figure 6).

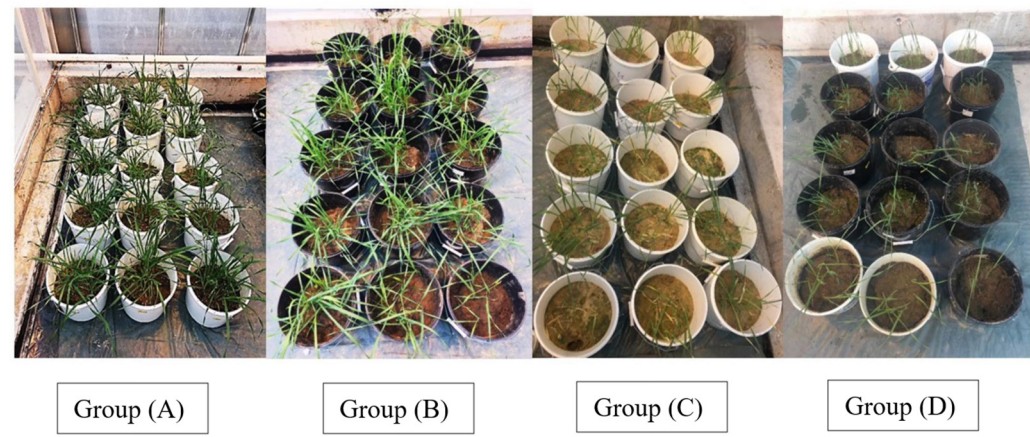

<div align="center">

| Group (A) | Group (B) | Group (C) | Group (D) |

</div>

**Figure 5.** Barley's green biomass after the application of urea and coated urea potassium humate in different soil types with different textures: loamy (**A**), sandy loam (**B**), sandy (**C**), and sandy clay (**D**).

**Figure 6.** The effect of urea (U100 and U200) and coated urea potassium humate (UPH100 and UPH200) on the green barley biomasses in four soil types with different soil textures, Arenosol (sandy), Cambisol (sandy loam), Luvisol (loam), and Retisol (sandy clay). The columns with the same letter are not significantly different at the 5% level of Duncan's test.

In the Retisol with sandy clay texture, there were no differences between the biomass under the treatments of U100 and UPH100 in comparison to the control. However, UPH200 recorded the highest significant biomass with an average of 23.32 g followed by U200 with approximately 42% less biomass at 13.45 g.

Moreover, the green biomass was increased significantly ($p \leq 0.05$) by N application in the Cambisol (sandy loam). The U200 and U100 treatments had the highest significant effect on biomass with averages of 45.04 g and 39.66 g, respectively. UPH200 and UPH100 did not affect the biomass compared to one another and control (Figure 6).

The fourth soil texture type is loamy, which represented the Luvisol even though its treatments recorded the highest values of green biomass of barley compared to the other soil texture types. However, the treatment had no statistically significant effect on the barley biomass. U200 recorded the highest significant average of 51.8 g followed by UPH200 at 49.7 g as shown in Figure 6. Furthermore, the soil and texture types had a significant ($p \leq 0.05$) impact on the dry matter of barley. The sequence of average dry matter values for each respective soil texture type was, as such, sandy loam (Cambisol) with the highest dry matter at 75.27 g followed by sandy clay (Retisol) at 71.55 g. The loamy (Luvisol) and sandy (Arenosol) soil recorded lower dry matter at 71.03 g and 70.15 g, respectively. On the other hand, the treatments showed no significant effects on the dry matter. The average dry matter values for each treatment, in sequential order, are as follows: U200: 74.96 g followed by UPH200: 73.1 g, UPH100: 71.42 g, U100: 70.84 g, and the control: 69.68 g.

## 4. Discussion

Observing the impact of both conventional urea and coated urea potassium humate on the dynamics of soil mineral nitrogen release in various soil with different texture types, we reported that the fertilization treatments significantly ($p \leq 0.01$) generated different release patterns of mineral nitrogen and its forms ($NH_4^+$–N and $NO_3^-$–N) as shown in Figure 4.

Regarding the fertilization treatments, the control recorded the lowest concentration of mineral nitrogen and its forms among all the treatments. Urea treatments obtained a significantly higher concentration of ammonium than coated urea potassium humate treatments in all soil and texture types in the following sequences U200 > UPH200 and U100 > UPH100. The explanation for the above is due to urea's rapid hydrolysis into ammonia ($NH_3$) and ammonium ($NH_4$) forms once applied in the soil [70,71] and an intensive nitrification process started so early [70,72]. The other reason is that coating urea with potassium humate caused a delay of 1–4 days in the peak of ammonium in various soils [12,22,37]. Potassium humate could absorb the ammonium due to the extensive porous surface area and the high cation exchange capacity [73]. Moreover, the reduced $NH_4^+$–N concentration observed as a result of the utilization of coated urea potassium humate treatments could potentially be attributed to the existence of a substantial quantity of acidic functional groups provided by the potassium humate [13,34,74].

This rapid release and the high concentration of ammonium and nitrate caused by urea treatments can be the main reasons for nitrogen loss via volatilization and leaching [75,76]. Therefore, coated urea potassium humate can provide a more enduring process of nitrogen release as it flattens the peak $NH_4^+$–N concentration, which would be expected to slow the nitrification and denitrification processes, allows more mineral N uptake by the plant, and leads to reduced N losses [22,37,47,77].

Among the four soil texture types, the mineral nitrogen and its forms ($NH_4^+$–N and $NO_3^-$–N) showed different patterns of release based on the variation in soil characteristics [8,70,78–80]. That implies that the chemical–physical traits of the soil that receives the fertilizer can significantly impact the fertilizers' efficacy and release rate [6,81]. It has been observed that Arenosol (sandy) exhibits the lowest concentration of mineral nitrogen right from the outset and this trend persists even after the application of fertilizers with low availability and low variability based on %CV [68]. Probably, the low content of nitrogen in Arenosol is because it is a light soil (i.e., low cation exchange capacity) [70,78,81]. The

peak of ammonium was recorded in the Arenosol, which is represented by a sandy texture after 7 days. Relative to the other soil types such as Cambisol and Luvisol, the peak of ammonium was delayed in Arenosol. The explanation for the delayed peak of ammonium in sandy soil (Arenosol) may be that the remainder of ammonium was consumed through processes of nitrification and leaching [78,79], or owing to the absence of microbial activity in this soil texture type [73,78,80]. The concentration of ammonium represented approximately 25% of the mineral nitrogen content in Arenosols.

Furthermore, Arenosol recorded the lowest content of nitrate, possibly due to low microbial activity in this soil type. The limited microbial activity in this soil type is due to its low organic matter content as presented in Table 1 [78]. This phenomenon can be attributed to the high potential for leaching losses of nitrate [78,81]. Notably, nitrate ($NO_3^-$), which is a negatively charged ion, is not retained by soil particles and is prone to leaching through the soil. As sandy soils have a relatively lower water-holding capacity, nitrates tend to leach at a faster rate as compared to soils with finer textures [78].

In Retisols (sandy clay), the low pH (Table 4) played an essential role in delaying urea hydrolysis due to inhibiting the urease enzyme activity [10,13,70]. However, The sandy clay (Retisol) soil obtained a higher significant concentration of mineral nitrogen and both forms of ammonium and nitrate compared to the other soil types. Due to the low pH, the peak of ammonium was delayed and recorded after 7 days, and at the time of the ammonium peak it comprised 44.4% of the mineral nitrogen content. Moreover, the highest significant concentration of nitrate was observed in Retisols with sandy loam texture.

There were no significant differences in mineral nitrogen concentration and its forms between the Cambisol (sandy loam) and Luvisol (loamy) (Table 5). The release was very rapid in Cambisol as a sandy loam textured soil. Therefore, the peak of $NH_4^+$–N derived from the fertilizer treatments and present in the soil was obtained at the beginning of the second day of the experiment. By contrast, the Luvisol present in a loamy texture obtained the peak after 4 days. The ammonium concentration represented approximately 30.7% and 33.3% of the mineral nitrogen content in the Luvisol and the Cambisol, respectively.

These soil types with a higher mineral nitrogen content are located in central and western parts of Lithuania, which are more fertile than the soils of the eastern part [81,82]. These findings were similar to a previous study on the mineral nitrogen content in the Lithuanian agricultural lands in 2014 [81]. The study revealed that sandy soils had the lowest levels of Nmin, while loamy sand and sandy loam soils had average levels. On the other hand, medium loam, clay loam, and clay soils had high levels of Nmin [81]. The explanation is that heavier textured soils contain larger amounts of mineral nitrogen with high availability [81,83].

The dynamics accumulation of mineral nitrogen and nitrate ($NO_3^-$) increased approximately linearly starting 1–2 weeks after application in all fertilizers treatments (Table 2). By contrast, ammonium concentrations increased rapidly to a peak 2–7 days after application followed by a decrease as the ammonium-N nitrified to nitrate, which means the nitrification process started at a very rapid rate [6]. The explanation is that, as time progresses, the levels of mineral nitrogen and its forms ($NH_4^+$–N and $NO_3^-$–N) can increase due to the disintegration of fertilizer compounds [1,43,73]. However, the release rate possibly does not correlate in a linear relation with the time because that process goes according to the needs of the plants for nutrients and their uptake [73,84]. The crucial role of microbial activity in the soil cannot be overlooked in this process [1,73].

The ammonium demonstrated a peak net accumulation after 28 days of the experiment, and nitrate exhibited persistent net accumulation until the conclusion of the experiment, owing to the continued presence of exchangeable $NH_4^+$–N in the soil as illustrated in Figure 4 [6]. What is more, the observed increase in the accumulation of $NO_3^-$–N toward the end of the experiment suggests a higher possibility for nitrification and, generally, for nitrogen mineralization and transformation [6].

Additionally, the barley's green biomass was influenced significantly ($p \leq 0.05$) by the application of urea and coated urea potassium humate treatments. The treatment of

UPH200 recorded the highest green biomass followed by U200 > U100 > UPH100, respectively. As illustrated in Figure 6, there were significant differences among the treatments within the soil type groups. Luvisol (loamy) under the treatments of urea and coated urea potassium humate obtained the highest biomass compared to the other soil types even though its accumulative mineral nitrogen during the investigation was less than the Cambisol (sandy loam) and Retisol (sandy clay) soils. There were no significant differences among the fertilization treatments for the barley biomass observed in Luvisol (Figure 6). The increase in the green biomass in Luvisol can be attributed to the nitrate concentration exhibiting a greater magnitude in comparison to the ammonium concentration within the loamy soil. Consequently, this led to a decreased loss of nitrogen and an increase in plant uptake, ultimately resulting in a higher yield of green biomass [85].

The Cambisol (sandy loam) exhibited the second highest values of the green biomass under the application of urea and coated urea potassium humate with notable ($p \leq 0.05$) variations among its fertilization treatments as depicted in Figure 6. The application of urea in Cambisol resulted in higher green biomass in comparison to the treatments of coated urea potassium humate. U200 showed a significantly higher biomass in comparison to UPH200 and UPH100, although no significant differences were observed between U100 and the other treatments. Cambisol is an agricultural land with good levels of nitrogen and neutral pH soil [81]. Therefore, the application of urea and coated urea took the normal path of release which leads to good productivity [86].

The Arenosol (sandy soil) impacted the green biomass of the barley significantly ($p \leq 0.01$). However, it is notable that the sandy soil displayed the lowest accumulative mineral nitrogen levels across all treatments. The treatments of coated urea potassium humate exhibited a significantly higher biomass of barley, especially with a high nitrogen application rate at 200 kg·N·ha$^{-1}$ followed by U200 > UPH100 with the same mean. Interestingly, the application of coated urea potassium humate proved to be more efficacious in promoting the highest significant biomass yield, as compared to both the urea and control treatments in Arenosol. This finding indicates that the implementation of coated urea potassium humate has the potential to enhance fertilizer use efficiency while concurrently alleviating N loss [46]. This is primarily due to the reduction in the rate of urea hydrolysis and N fertilizer conversion in the soil [45,46,87].

On the other hand, the sandy clay soil (Retisol) recorded the lowest values for the barley biomass. The coated urea potassium humate recorded the highest significant biomass followed by urea with the same application rate at 200 kg·N·ha$^{-1}$. The unimpressive barley biomass yielded in the sandy clay soil, despite it exhibiting the highest significant concentration of mineral nitrogen. The explanation is found in the sensitivity to aluminium toxicity, which is caused by very high levels of aluminium dissolved in the soil water [88]. It is one of the factors that lead to a decrease in barley yields in acid soils [88,89]. This mineral is particularly detrimental to barley due to its high sensitivity toward it [88].

## 5. Conclusions

Controlled-release fertilizers such as urea coated with potassium humate have the capacity to enhance the levels of soil mineral nitrogen. This includes the fundamental forms (ammonium $NH_4^+$–N and nitrate $NO_3^-$–N) content, thereby fulfilling the nitrogen necessities for the growth of barley. The potassium humate as a coating cover can be a beneficial instrument to mitigate the risk of N loss by delaying the release of nitrogen in ammonium form for 1–4 days to mitigate nitrogen losses by leaching or volatilization which is caused by the rapid release of urea. However, the efficacy of such fertilizers is significantly reliant on their chemical properties and application rate, as well as the chemical–physical properties of the soil in which they are applied. The addition of different fertilizer materials elicited varied responses from the soils. The coated urea potassium humate granules proved to be more efficient in soils such as Arenosol (sandy) and Retisol (sandy clay) than the Cambisol (sandy loam) and Luvisol (loam) in terms of maintaining optimal nitrogen levels and reducing the risk of nitrogen losses.

**Author Contributions:** Conceptualization, R.M. and S.S.; methodology, R.M., J.V. and S.S.; formal analysis, R.M.; investigation, S.S.; data curation, S.S.; writing—original draft preparation, S.S.; writing—review and editing, R.M. and J.V.; visualization, S.S.; supervision, R.M. All authors have read and agreed to the published version of the manuscript.

**Funding:** This research received no external funding.

**Institutional Review Board Statement:** Not applicable.

**Informed Consent Statement:** Not applicable.

**Data Availability Statement:** The datasets generated and analyzed during the current study are available from the corresponding author upon reasonable request.

**Acknowledgments:** The authors would like to thank everyone from the Rumokai, Dotnuva, and Vezaiciai experimental stations of the Lithuanian Research Centre for Agriculture and Forestry for providing help during the experiment, including sampling collection. We also express our gratitude to AB Achema Scientific Experimental Laboratory, Lithuania, for providing us with the fertilizer materials.

**Conflicts of Interest:** The authors declare that they have no known competing financial interest or personal relationships that could have appeared to influence the work reported in this paper.

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
