# Peer review of "Mineral Nitrogen Release Patterns in Various Soil and Texture Types and the Impact of Urea and Coated Urea Potassium Humate on Barley Biomass"

_soilsystems, doi:10.3390/soilsystems7040102_

Round 1

Reviewer 1 Report

Comments and Suggestions for Authors

Dear Editor and authors, the article has the potential to be published but needs modifications as follows:

-In the introduction and discussion, authors need to segment the text into paragraphs.

-On lines 67-68, the authors say that humic substances are macromolecules, but the most accepted theory is that humic substances are a supramolecular structure. I think it is valid for the authors to change the term, or explain these two theories of formation of humic substances.

- In the introduction, the authors also need to highlight the effect and effectiveness of the application of products containing humic substances, highlighting which main properties and how they influence these responses.

- More information on the humic substances used should be provided, as their effect is highly dependent on their properties, for example, humic substances derived from organic waste through composting, also have a biostimulant effect.

- The results section needs to be redone, there is nothing in the figures that shows whether or not there was a statistical difference, in addition this information needs to be included in the caption.

- The correlation in table 2 is unnecessary and explains the data, what explains the data is the effect on time observed in the previous figures, there is an error in the approach to the correlation analysis

- In table 3 and 4, in relation to the soils, the effect must be compared by the soil and not by the texture, because together with the texture it varies other soil properties and not only it, example: mineralogy, soil organic matter content.

- It would be interesting to correlate production data with variables related to soil N dynamics.

- The discussion needs to be redone based on what was requested in the results section.

Author Response

Thanks for your comments, all of them are useful, and I will consider all of them to improve my manuscript as much as I can. Please find the response letter uploaded.

Reviewer 2 Report

Comments and Suggestions for Authors

In this manuscript, the authors investigated the process of mineral nitrogen release in different textured soils to understand the interaction between soil characteristics and mineral nitrogen release and evaluate the impact of conventional urea compared to coated urea potassium humate on barley biomass production using a pot culture study. The study objective is novel and findings reported in this manuscript will advance the existing knowledge on N mineralization. However, I have some observations indicated below which are to be addressed by the authors.

1.           Nitrogen (N) mineralization is a pretty long process which could be understandable if it is continued for 90-100 days or more. However, the authors studied the N mineralization process only for 45 days. So, I am not sure how much the study is effective. Anyway, I appreciate the outcomes of this study. The authors can consult with the following articles for reference.

i. Islam, M.R., Bilkis, S., Hoque, T.S., Uddin, S., Jahiruddin, M., Rahman, M.M., Rahman, M.M., Alhomrani, M., Gaber, A. and Hossain, M.A. 2021. Mineralization of farm manures and slurries for successive release of carbon and nitrogen in incubated soils varying in moisture status under controlled laboratory conditions. Agriculture 2021, 11(9), 846; https://doi.org/10.3390/agriculture11090846.

ii. Islam, M.R., Bilkis, S., Hoque, T.S., Uddin, S., Jahiruddin, M., Rahman, M.M., Siddique, A.B., Hossain, M.A., Marfo, T.D., Danish, S. and Datta, R. 2021. Mineralization of farm manures and slurries under aerobic and anaerobic conditions for subsequent release of phosphorus and sulphur in soil. Sustainability 2021, 13(15), 8605; https://doi.org/10.3390/su13158605.

2.           The authors used the terms mineral nitrogen, NH4-N and NO3-N which are confusing. Because NH4-N and NO3-N are also mineral N. So, it is better to use total N, NH4-N and NO3-N.

3.           Line 119-134, fonts have been changed, should be checked.

Author Response

(The authors gave the same response as above.)

Reviewer 3 Report

Comments and Suggestions for Authors

The manuscript addresses the effect of controlled-release urea on different types of soils. The topic is of great importance to the scientific community and falls within the scope of the journal. However, the way it is presented does not contribute to scientific advancement in the area.

The title does not agree with the manuscript, no mineralogical study, referring to the types of soil studies, was added to the work to justify the title of the manuscript.

The Summary is poor, without any consistent results. There are serious signs of methodological errors, or at least, poor description of the Material and Methods that compromise the reliability of the work. Basically, the authors describe the forms of nitrogen (nitrate and ammonium) in the soil after applying the treatments. However, the way in which soil samples were collected during the experiment was not described.

There is a very large variation in the pH of the soils evaluated, this can influence the treatments and especially the development of the test crop. But the authors do not present any comments on this matter.

Given the above, I do not feel comfortable recommending the work for publication.

Author Response

(The authors gave the same response as above.)

Reviewer 4 Report

Comments and Suggestions for Authors

Dear authors,

I have some comments that need to be taken into account.

General comments:

The graphical abstract is not readable. Increase please the font size. There are some typos in the manuscript and the reference list. Please check that all fonts are the same size.  Figures 2 and 3 are not necessary, Instead, a scheme of the experimental design should be better. In Figure 3, what is the green arrow?

The different patterns of nitrogen release should. be further discussed (in relation to texture but also to the basic soil characteristics from Table 1 etc., the pattern of NH4+ in the sandy clay with its several release peaks,…).

Specific comments:

L49: it is

L53: “brilliant means” is too familiar

Materials and Methods

There is something with the font size here.

A map with the study sites should be nice even in an annexe since it is a pot experiment.

L155-156: Samples were procured… what do you mean, reformulate, please? Explain how (sampling) what sample size (g) was needed for the analyses.

L157: was there a reason for these days? Barley growth?

Results:

The order of ABCD in Fig. 4-6 is strange!

In Table 2, I think you can also add sand, loam, etc near the WRB soil type.

Where is the table with the barley biomasses? Or dry matter

Fig. 8, (gm) is for gram> if so it is (g)

Discussion:

I think the discussion should underline/develop more the differences obtained.

L346-351: reformulate

L354-355: some reference here

L364-365: Why? The dynamic of the release could be explained/discussed in more detail.

L425-428: why?

References:

There are some typos here and some format problems.

Comments on the Quality of English Language

The English need some minor revisions but it is quite acceptable.

Author Response

(The authors gave the same response as above.)

Reviewer 5 Report

Comments and Suggestions for Authors

Comments on “Mineral Nitrogen Release Patterns in Various Soil and Texture Types and Impact of Urea and Coated Urea Potassium Humate on Barley Biomass”

This paper has several methodological omissions, and my recommendation is to reject it in its current form, but reconsider after major revision. There are sufficient data that it could be comprehensively rewritten and resubmitted, but the time required would be much greater than the time normally allowed by MDPI to respond to reviewer criticisms. Major criticisms:

1.       Mineral nitrogen concentrations in some soils much higher than would be expected from the amount of urea applied. For example in the Sandy Clay soil by day 45 the mineral N concentration was about 500 mg/kg (Fig 6C), whereas what would be expected from 100 kg N with a bulk density of 1 g/cm3 mixed evenly through the 0-10cm layer is 100 mg/kg plus the value in the Control (~100 mg/kg), so 200 mg/kg would be expected. This could be a valid value if the bulk density were 0.25 g/cm3, but there are few soils like this. Bulk densities are not presented, but these are essential for converting between mg/kg and N recovery.

2.       Pots were allowed to drain from the bottom, removing nitrate in a way that was not measured. Normally in such experiments pots are lined with a plastic bag and watered to weight so leaching doesn’t need to be measured.

3.       Barley chosen as the test plant even though one of the soils had a pH of 4.5. This is too acid for barley, causing poor growth (Fig 7). This acidity constraint limits the usefulness of barley as a test plant, and its response to nitrogen. In a field situation the field would either be limed, or planted to a more acid-tolerant crop such as rye or triticale. Either of these species would have been a more suitable test crop.

4.       N concentration was not measured on the harvested barley material. This would have enabled calculation of N uptake, and calculation of the fate of applied N into soil ammonium, soil nitrate and plant forms.

5.       The statistical analysis has mixed incompatible errors. In Table 4 the coefficient of variation for mineral N was 23% for the sand compared with 172% for the sandy clay. This violates one of the assumptions of ANOVA, that errors are similar across treatments. Some specialist statistical procedures take this into account, but there is no mention that this was done in the Methods. The simplest approach is to undertake a separate ANOVA for each soil.

6.       The writing style has extremely long paragraphs. A paragraph should be focussed around one idea. It should start with an introductory sentence and finish with a concluding sentence that brings the logic together. A good Discussion should start with a paragraph about whether the initial hypothesis is supported by the data. In this case it is whether humate coated urea delayed the release of urea-N into first ammonium then nitrate pools. It should then weave in qualifiers of how the release pattern differed with soil type, and other evidence that supports this from the literature. Subsequent paragraphs should address lesser points that can be drawn from the data.

7.       The statistical analysis doesn’t address the key hypothesis that the coating delays release of the urea-N. I suggest the analysis reported in Table 2 be analysed with a linear mixed model of Fertiliser * Days for each soil, and test the significance of differences in slope between U100 vs UPH100, and U200 vs UPH200. The data presented in this Table don’t include the slope, only the R2 which is only a measure of its conformity to a non-horizontal straight line. SPSS has linear mixed models.

8.       Figs 4-6 should have error bars. This avoids readers attempting to get meaning from every little bump in the chart.

Where to next? I suggest the senior author spend time looking through resources on the internet on writing scientific papers. It’s much easier to write knowing the rules for the style.

Secondly I suggest following a “model paper” that’s already published and similar to the data presented here. There is likely to be a suitable paper in the reference list. This will provide a guide on the writing style used in other similar studies.

Thirdly choose which points are worthwhile from the current Results and Discussion sections. Results should refer the reader to the appropriate Figure or Table and draw out the points that are later taken up in Discussion and Conclusions. This should only be 1-2 sentences per Figure or Table as readers can gain the information more easily from these than from text.

Minor points

Title “…Urea and Potassium Humate Coated Urea”

line 20 …to potassium humate coated urea

Line 22 …including a no fertilizer control

Lines 26-27 It is disappointing to readers to be told the treatment “significantly affected” without describing how it was affected. For example “Ammonium concentrations peaked in all soils within 5 days of application, while nitrate concentrations increased at an approximately linear rate over the 45 days of the experiment. Concentrations of both N forms were highest on the sandy clay and lowest on the sand. Potassium humate delayed the peak of soil ammonium by x days and its conversion into the nitrate form was x-y% slower than with untreated urea.”

Lines 39-40 and accounts for approximately 43% of global nitrogen fertilizer sales

Line 45 loss culd cause ecological

Line 47 up to 143 kg N/ha [Note that leaching is minimal in semi-arid areas where urea is used as fertiliser]

Line 49 Therefore it is essential to improve the efficiency of urea fertiliser to reduce nitrogen loss

Line 50 fertilizer application

Line 53 is one strategy

Line 54 Controlled-release

Line 55 N release pattern

Line 65 the use of organic

Line 73 mitigate losses

Line 74 enhances nitrogen in the ammonium form NH4 rather than the ammonia form NH3, which would be expected to reduce environmental damage

Line 77 groups in the humic compounds with urea

Lines 92-94  [51]. Coating of urea with humate leads to a stable product [37] with controlled nitrogen release [37-39] and the advantageous

Line 96 Different soils

Line 100 urea coated with

Lines 101-103 By quantifying these aspects, we can further our understanding of nitrogen dynamics in different soil types following application with uncoated and coated urea. [The author may be able to tighten this further into a hypothesis about the coating delaying the release of urea]

Line 110-112 Potassium Humate (AB Achema, Jonavos, Lithuania).

Line 145 with 10kg of air-dried soil packed to a bulk density of approximately x t/m3. [worthwhile to include bulk density as it is needed to convert from mg/kg to kg/ha or total recovery]

Line 149 with the 4 soil

Line 156 from each pot

Line 155-156 Soils were sampled to a depth of 10cm from each pot

Line 158 “at equal intervals” needs better explanation

Fig 2 & 3 Unnecessary so remove

Line 163 Soil for the pot experiment was obtained at depths of 0-30cm from each of four soil types (Table 1). The soil was air-dried…

Line 168-169 Plant-available

Line 174 nitrate concentration

Line 174-175  ammonium concentration

Line 175 was calculated

Line 178-180 Barley vegetative material was cut 45 days after sowing and dried at 105C.

Fig 4 Label as Sand, Sandy loam etc rather than A B C. Analyse data by a suitable curve in SPSS regression and show an error bar. An error bar could be calculated by ANOVA. Curve fitting will indicate whether there has been a different time to peak for the various fertiliser materials and soils.

Subsequent sections are too wordy.

There should be discussion on N recoveries (N applied vs N recovered in mineral N and plant uptake), and why mineral N levels were so low in the sand. Had it been moved to below the 0-10cm layer that was sampled?

Table  3 No evidence here that coating has affected nitrate or ammonium at 45 days, although there may have been difference earlier.

These values appear to exceed 100% recovery, as these are average values from the 0-30cm depth. Was there a priming effect of added N stimulating release of N from organic matter? Why as the Sand so low ? Had the N leached out through over-watering? In this stats analysis was there a Soil x Fertiliser interaction?

Comments on the Quality of English Language

Some minor wording changes are suggested. The Results and Discussion sections are too wordy and the Discussion is not arranged into paragraphs. These need a complete rewrite into about one third their current length.

Author Response

(The authors gave the same response as above.)

Round 2

Reviewer 1 Report

Comments and Suggestions for Authors

Dear Editor and authors, the manuscript was improved and can be accepted, but the quality of figure 4 is poor. The authors should improve the quality of this in the published version.

Author Response

Thanks so much for your comments.
The authors will try to do their best as much as possible.

Reviewer 4 Report

Comments and Suggestions for Authors

Dear authors,

I am satisfied with most of the replies/changes however, there are still few issues.

The language in the new sentences can be improved. Many typos and format errors (probably due to the display of markups.

 L60: “brilliant means” This is an informal/familiar formuling. You can’t use that in a paper. Please reformulate.

           L157: was there a reason for these days? Barley growth?

Do you mean the sequences of the days of the sampling from the pots? It’s based on a pre-study to study the release of these materials in incubation tests without plants.

>>> This info should be given in the manuscript. (L174)

            Fig. 8, (gm) is for gram> if so it is (g) ..  both appriviation are used and correct.

>>> not right. Please change to (g)

The only unit symbol for gram that is recognised by the International System of Units (SI) is "g" following the numeric value with a space.

L301-302: reformulate (These soil types…)

L397-398: reformulate

L425-428: why?

-          Because there are many factors that could affect biomass productivity not only the uptake of nitrogen.

Then say so in the text and explain. Now L480-485 I suppose.

L498: Therefore instead of “so”

Comments on the Quality of English Language

The English language was improved but there are still errors (especially in the new parts).

Author Response

Thanks for your efforts and comments!

Please, find my responses in the uploaded cover letter.

Reviewer 5 Report

Comments and Suggestions for Authors

This version is a significant improvement on the previous version, but the “Results” section is too long at 3 ½ pages of text. This section is the most challenging to write in an interesting way, so the answer is to write it as briefly as possible. The text should do no more than refer the reader to the appropriate Figure or Table and describe the points that are later taken up in Discussion and Conclusions. The important findings of each Table of Figure should be described in no more than 1-2 sentences each, and leave the reader to find more detail in these Tables and Figures.  I suggest the target length of text for this section is less than a single page.  

The “Discussion” section at 2 ½ pages is also too long relative to the data presented and poorly focussed. It should be structured around the objectives in lines 115-117. The first paragraph should address objective 1 of evaluating the different patterns of mineral N release, and the second paragraph objective 2 of appraising the impact on barley biomass. There should also be a paragraph on how the humic coating affected the release pattern. A paragraph in Discussion should start with a sentence based on observation from the experiment, the following sentences refer to literature, or limitations of the techniques, and the final sentence is a conclusion or implication. It should not have its opening sentence based on with literature, like a literature review. I suggest the target length for the Discussion of 1 ½ to at most 2 pages.

Another limitation in the study is that it is attempting to measure the rate of release of urea, but there were no direct measurements of release, only the accumulation of N in ammonium and nitrate forms. These forms are dynamic, with N taken up by plants, and lost to leaching to below the 10cm layer. We can only infer the release pattern. This should be brought out in the text.

Minor points

Line 30 lowest accumulation of mineral

Line 31 which at peak accounted

Line 32 25-44% [doesn’t need decimal point in the abstract]

Line 33 Describe the significant effects, eg caused a 12-50% increase in barley biomass relative to an unfertilised control.

Line 43 barley

Line 47 its high nitrogen

Line 49 the efficiency

Lines 54-57 leaching and by adding to the emission of greenhouse gases into the atmosphere, with reports of up to 160 kg N/ha/yr lost to leaching and up to 143 kg N/ha/yr lost as greenhouse gases.

Line 60 is one strategy to

Line 65-67 Delete as sentence adds no value

Line 80-81 which demonstrate chelating

Line 83 Coating urea with humic acid

Line 84 ammonification and nitrification [use the correct sequence order]

Line 85 conversion of urea-N I slowed due to

Line 86 This is due

Line 91 transformation to the ammonium form (NH4-N) rather than the ammonia form (NH3)

Line 96-97 can slow the release of urea into the ammonium form, which then also slows its conversion into the nitrate form, leading to lower N losses.

Line 111 delete “and the…methodology” as it adds no value

Line 124 urea coated with

Line 127 urea coated with

Line 153 add “Note that analytical methods are described in section 2.4”

Line 160 urea coated with

Line 162 in 4 groups, one for each soil type, and each group consisted for 15 pots

Line 163 was 25 cm in diameter and 25 cm in height, and filled

Line 168 urea coated with

Line 184 through a 2mm sieve

Line 208 nitrogen in the top 10cm of the pots (NH4-N and NO3-N)

Line 212 the Arenosol (sandy) showed the lowest peak of (N4-N), where the Retisol (sandy clay) had the highest peak.

Line 230 the concentration of

Lines 236-241 are a repeat of lines 208-213

Line 232 in the Luvosol

Line 272 Concentration of ammonium (NH4-N), nitrate (NO3-N) and total mineral N in the 0-10cm layer of the pot experiment.

Line 280 The accumulation of mineral N in the top 10cm of the soil over the 45 days

Line 284 By contrast the highest

Line 289 the U200

Line 291-292 U200 signficiantly increased…and nitrate by day 45? At 94.84

Line 294 By contrast

Line 301 Remove “Moreover”

Line 306 as neutral

Line 309 increased the pH

Line 312 However, the application of U200 increased the pH

Line 314 as delaying the

Line 318 values of the 0-10cm layer of the pots after

Line 372was increased significantly (p < 0.05) by N application in the Cambisol

Line 401 “did not correlate with time” is incorrect. It just did not follow a linear relationship with time, but instead increased then decreased. An appropriate curve was not fitted.

Line 461 is prone to leaching through the soil

Lines 466-468 There was no significant differe in mineral N concentration between the sandy loam (Cambisol)  and loamy soil (Luvisol)

Line 474 explanation is that

Line 478-479 recorded the highest biomass followed by U200>U100>UPH100 respectively.

Line 483 than the sandy

Line 486-487 nitrate rather than the ammonium concentration

Line 490 (Cambisol) exhibited

Line 494-495 biomass that UPH200

Line 502 exhibited significantly higher biomass

Line 520 sensitivity to aluminium

Lines 539-540 (sandy clay) than the Cambisol (sandy loam), Luvisol (loam)…levels and reducing the risk of losses.

Comments on the Quality of English Language

Comments about improving English are included in the section above. The main issue is excessive length of the Results and Discussion sections, and their poor organisation. Text in the Results section should describe each Figure or Table in 1-2 sentences each drawing out the points that are later taken up in Discussion.

The Discussion should address the objectives in Introduction and follow the same structure. There is no clear discussion of whether the coating delays release of urea, other than a comment that it is by 1-4 days. This hardly justifies the extensive literature in the Introduction about coatings on urea.

Author Response

(The authors gave the same response as above.)

Round 3

Reviewer 5 Report

Comments and Suggestions for Authors

Comments on “Mineral Nitrogen Release Patterns in Various Soil and Texture Types and Impact of Urea and Coated Urea Potassium Humate on Barley Biomass” Version 3

Most sections of this manuscript read well and are of a length suitable the data presented. However, the Results section is still too long. It describes in detail findings that are already presented more clearly in Figures and Tables. The text should instead describe features of the data that are later taken up in the Discussion and Conclusions. Some rewording suggestions are included below that could greatly reduce the Results section. Please check whether points covered in the Discussion are still covered in the reworded Results section.

The paper would be easier to read if the soil classifications are mentioned once (eg Arenosol), thereafter refer to the soils only by their texture. This would be easier to understand for soil scientists not familiar with the European classification system. The experiments were only conducted on repacked topsoil, whereas the classification refers to the full soil solum.

Line 28 The Retisol

Line 49 However, the

Line 58 fertilizer application

Lines 63-64 advance in resolving

Lines 91-94 Delete sentence “The interplay…humate”  as unnecessary

Line 95 urea into the

Line 98 thus decrease

Line 111 urea-potassium

Line 130 soils. Soil profiles

Line 155 days by growing

Line 168 collected to a depth of

Line 178 Soil for the experiment was collected from

Line 193 Delete “The weight…barley”

Line 205-227 The concentration of ammonium-N in the top 10cm of soil in the pots increased rapidly after urea application, reaching a peak after 1-7 days, followed by a gradual decline to near-zero levels in all soils except the sandy clay (Figure 4). The peak ammonium concentration varied with soil, urea source and rate, and was highest on the sandy clay, followed in sequential order by the sandy loam, loam and sand. The highest peak concentrations were in the U200 and UPH200 treatments. However, there was no significant linear relationship between ammonium-N concentration and time.

Line 228-265 By contrast the nitrate-N concentration increased at a linear rate over the 45 days of measurements (Fig 4, Table 2). The highest peak was found in the sandy clay, followed by the loam, sandy loam and sand.  Total mineral N also increased with time, but only achieved significance at the 5% level in 11 of the 16 soil-fertiliser combinations tested (Fig 4, Table 2).

Line 275-290 At the conclusion of the experiment on day 45, soil in the pots to a depth of 30 cm was analysed mineral N.  Among soils the highest mineral N concentrations were in the sandy clay, followed by the loam and sandy loam, while the lowest concentrations were in the sand (Table 3). Among fertiliser treatments the highest concentration was in U200 followed by UPH200, UPH100 and U100.

Line 338-339 Germination of the barley seed commenced

Line 350-382 There were significant responses to applied nitrogen in barley biomass (Figure 6). The highest yields were achieved on the loam and sandy loam soils, with much lower yields on the sandy clay and sandy soils (Figure 6). Averaged across all soils the highest biomass was in the U200 treatment (75g), followed in sequential order by UPH200 (73.1g), UPH 100 (71.4g), U100 (70.8) and Control (69.7).

Line 398-399 hydrolysis into ammonia (NH3) and ammonium (NH4+) forms once

Line 410-412 release as it flattens the peak NH4+-N concentration, which would be expected to slow the nitrification and denitrification processes, allow more mineral N uptake by the plant, and lead to reduced N losses.

Line 421 Relative to

Line 428-429 lowest concentration of nitrate, possibly due to low microbial.

Line 439-440 7 days and at the time of the ammonium peak it comprised 44.4% of the mineral nitrogen.

Line 440 significant concentration of

Line 446 While the Luvisol

Line 448 33.3% of mineral nitrogen in the

Line 458-463 The dynamic accumulation…nitrate (NO3-) and mineral N increased approximately linearly starting 1-2 weeks after application in all treatments (Table 2). By contrast ammonium concentrations increased rapidly to a peak 2-7 days after application followed by a decrease as the ammonium-N nitrified to nitrate.

Line 499 The Arenosol

Table 2 Add column headers for NO3 and Nmin “mg/kg”, and “Correlation coefficient” immediately above the number values as the present layout is unclear

Table 3 caption add “at day 145”. Also the Nmin column should be on the far right for consistency with other Tables

Comments on the Quality of English Language

English expression is mostly good. Some suggestions are in the above box.

Author Response

(The authors gave the same response as above.)
